
# 1  Wall loss of semi-volatile organic compounds in a Teflon bag chamber
# 2  for the temperature range of 262-298 K

Longkun He[1], Wenli Liu[2], Yatai Li[1,3], Jixuan Wang[1], Mikinori Kuwata[2], Yingjun Liu[1]
[1]State Key Joint Laboratory of Environmental Simulation and Pollution Control, College of Environmental Sciences and
Engineering, Peking University, Beijing, 100871, China
[2]Department of Atmospheric and Oceanic Sciences and Laboratory for Climate and Ocean-Atmosphere Studies, School of
Physics, Peking University, Beijing 100871, China
[3]Now at College of Public Health, Zhengzhou University, Zhengzhou, 450001, China
*Correspondence to*: Mikinori Kuwata (kuwata@pku.edu.cn), Yingjun Liu (yingjun.liu@pku.edu.cn)
**Abstract.** Teflon bag chambers have long been used for investigating atmospheric chemical processes, including secondary
organic aerosol formation. Wall-loss process of gas-phase species in Teflon bag chambers has typically been investigated at
around room temperature. Recent laboratory studies started employing Teflon bag chambers at sub-273 K conditions for
simulating wintertime and upper tropospheric environments. However, temperature dependence in vapor wall-loss processes
of semi-volatile organic compounds (SVOCs) in a Teflon bag chamber has not well been investigated. In this study, we
experimentally investigated wall-loss process of $C_{14}$-$C_{19}$ *n*-alkanes in a 1 m$^3$ Teflon bag for the temperature range of 262 to
298 K. Enhanced wall losses of the tested *n*-alkanes were observed following the decrease in temperature. For instance, 65%
of $C_{14}$ *n*-alkane was lost to the wall 15 hours after injection at room temperature, while the corresponding value was 95% at
262 K. The experimental data were analyzed using the two-layer kinetic model, which considers both absorption of gas phase
species to the surface layer of Teflon wall and diffusion to the inner layer. The experimental data demonstrated that absorption
of gas phase species by the surface layer enhanced at lower temperature. The temperature dependence in absorption was well
accounted using the equilibrium dissolution model of organic compounds to the Teflon surface by considering reduced
saturation vapor pressure at lower temperature. On the contrary, diffusion process of *n*-alkanes from the surface to inner layer
slowed down at reduced temperature. Hence the relative importance of the surface and inner layers on wall-loss process
changes with temperature. Mechanistic studies on these processes will need to be conducted in the future to quantitatively
predict the influence of temperature-dependent wall-loss processes of SVOCs on laboratory experimental results.



## 1 Introduction

The environmental chamber is one of the most widely-used laboratory systems for studying chemical processes in the atmosphere, including formation of secondary organic aerosol (SOA) (Clark et al., 2016; Nakao et al., 2011; Ng et al., 2007; Song et al., 2005). The environmental chambers are typically made of Teflon films or stainless steel (Cocker et al., 2001; Bunz et al., 1996; Voigtlaeander et al., 2012). Existence of walls in the environmental chambers induces losses of both vapors and particles due to their deposition on wall surfaces (Mcmurry and Grosjean, 1985; Krechmer et al., 2020). Wall loss of gas-phase organic compounds in the environmental chambers can lead to underestimation of SOA mass yields. For instance, injection of seed particles into Teflon bag has been shown to increase SOA yields by a few times due to the reduced relative importance of the chamber wall as a condensation sink in the system (Kroll et al., 2007; Zhang et al., 2014).

Vapor wall loss in Teflon bag chambers, especially that for semi-volatile organic compounds (SVOCs), has been intensively investigated in the last decade (Matsunaga and Ziemann, 2010; Yeh and Ziemann, 2014; Yeh and Ziemann, 2015; Zhang et al., 2015; Krechmer et al., 2016; Huang et al., 2018b; Pratap et al., 2020; Yu et al., 2022). For instance, Matsunaga and Ziemann (2010) studied wall-loss process of alkanes, alkenes, alcohols, and ketones. These previous wall-loss experiments were dominantly conducted at around room temperature (293~303 K), as most of the chamber studies employed the corresponding temperature range (Hidy, 2019). The experimental results were often modeled by assuming equilibrium dissolution of the organic compounds into the Teflon film. A more recent study separately considered the surface and inner layer of the Teflon film for explaining the loss process more quantitatively (Huang et al., 2018b).

Recently, a growing number of environmental chamber experiments have been conducted at low temperatures to simulate wintertime/upper tropospheric conditions in laboratory (Huang et al., 2018a; Pratap et al., 2019; Quelever et al., 2019; Simon et al., 2020; Wang et al., 2022). For instance, some SOA formation experiments have been conducted for the temperature range down to 223 K using stainless steel chambers such as the Aerosol Interaction and Dynamics in the Atmosphere (AIDA) and Cosmics Leaving OUtdoor Droplets (CLOUD) chambers (Huang et al., 2018a; Simon et al., 2020). Teflon bag chambers have also been employed for the temperature range down to 258 K (Kristensen et al., 2017; Deng et al., 2021). These studies demonstrate that temperature is an important parameter determining both mass yields and chemical composition of SOA. Vapor wall loss of SVOCs in the environmental chambers for the corresponding temperature range needs to be understood for better interpreting these experimental data in a quantitative way. So far only one group attempted to investigate vapor wall loss below room temperature, by measuring the size evolution of levoglucasan particles injected into a Teflon chamber (Pratap et al., 2020). However, the experimental results were confounded by slow evaporation of levoglucosan from particles at low temperatures.

This study investigated vapor wall loss of $C_{14}$-$C_{19}$ n-alkanes in a Teflon chamber for the temperature range of 262 to 298 K by monitoring the evolution of their gas-phase concentrations following a pulse release. The experimental results were



analyzed using the two-layer kinetic model, which considers partitioning of gas phase SVOCs to the surface layer, as well as
further diffusion to the inner layer. Temperature effects on the two processes were evaluated separately.

## 2 Experimental

### 2.1 Teflon chamber experiments

Figure 1 shows the experimental setup. The experiment was conducted using a fluorinated ethylene propylene (FEP)
bag with the volume of 1 m$^3$. The thickness of the FEP film for the bag was 75 μm. The dimension of the bag was 260 cm ×
55 cm × 70 cm. The chamber volume was experimentally validated by employing $CO_2$ as a tracer (Figure S1). The bag was
newly purchased for the experiment, meaning that it was employed for no other experiments. The bag was installed in a chest
freezer (Type 2288, Nixue Inc.), which was equipped with an additional internal thermal insulation layer. Two fans were
installed in the freezer to promote the mixing of the air. The temperature of the freezer was measured at 3 points using
temperature sensors (Figure 1). Temporal variation of temperature was ± 0.5 K at 262 K.
Throughout the experiments, purified air was employed. The purified air was produced using a zero air generator
(Model 747−30, AADCO Instruments, Inc.) and further purified using a hydrocarbon trap (BHT-2, Agilent Technologies, Inc.).
Hydrocarbon concentration in the purified air was less than 5 ppbv. Relative humidity (RH) was less than 0.1%.
Solutions containing $C_{14}$ - $C_{19}$ $n$-alkanes (Konoscience Inc., > 98%) were prepared and injected into the chamber.
Hexane (Fisher Chemical Co., HPLC grade) was employed as the solvent. The purities and saturation vapor pressures of all
chemicals are given in Table S1. The solutions were injected to the chamber using a syringe pump (Fusion 200 Touch, Chemyx
Inc.) and a nebulizer (TR-30-A1, Meinhard Inc.) through polytetrafluoroethylene (PTFE) tubing, as shown in Figure 1. The
use of nebulizer expedited the evaporation of the solution.
Eight sets of wall-loss experiments were conducted in the temperature range of 262 to 298 K. Prior to each experiment,
the chamber was continuously flushed using purified air, until the concentration of investigated $n$-alkanes dropped to the
background level. To start an experiment, the chamber was switched to batch mode and the solution was injected to the
chamber at room temperature. The injection lasted for 13 mins, with a liquid flow rate of 100 μL min$^{-1}$. The air flow rate of
the nebulizer was 0.7 L min$^{-1}$. The resulting initial concentrations ($C_0$) of individual $n$-alkanes in the chamber ranged from 4
to 50 μg m$^{-3}$ assuming no wall loss. The solution used for low-temperature experiments (< 278 K) did not contain $C_{18}$ and $C_{19}$
$n$-alkanes to avoid formation of particles. For experiments below room temperature, the cooling system of the freezer was
turned on one hour after the completion of the injection. The operation procedure was employed to avoid homogeneous
nucleation and subsequent condensational growth of aerosol particles. Measurements using an optical particle counter (11-D,
GRIMM Aerosol Technik Ainring, Germany) experimentally confirmed negligible abundance of aerosol particles in the
chamber (< 0.5 μg m$^{-3}$). It took ~3 hours for the temperature in the freezer to drop to a stable level after injection (Figure S2).





Although the air in the bag leaked out during experiments due to compression of the bag by its own weight, absence of intrusion
of room air to the bag was confirmed by observing no changes in contaminant signals (Table S2).

Concentrations of SVOCs in the chamber were quantified using the semi-volatile thermal desorption aerosol gas

chromatograph (SV-TAG, Aerodyne Research Inc. & Aerosol Dynamic Inc., USA)  (Zhao et al., 2013). The gas
chromatography-mass spectrometer (GC-MS) (7890B, Agilent Technologies, Inc.) was employed for the system. Detailed
descriptions of the SV-TAG operation and performance tests were presented in our previous papers (Li et al., 2022a; Li et al.,
2022b). Herein, chamber air was sampled through ~1 m long perfluoroalkoxy alkane (PFA) tubing (1/4 inch in diameter). Prior
to sampling, the chamber air passed through the PFA tubing at 0.5 L min$^{-1}$ for at least 20 min for passivating the tubing wall
(Matsunaga and Ziemann, 2010). Samples were periodically collected for 5 min at 4 L min$^{-1}$ for each time at 1-15 hours after
injection. As the absence of particles was confirmed, only gas-phase SVOCs were sampled by the SV-TAG. The instrument
response to $n$-alkanes was calibrated with standards before and after each experiment (Figure S3), utilizing the in-situ automatic
injection system (Isaacman et al., 2011). The gas-phase concentrations of SVOCs were calculated from the measured quantity
of SVOCs and sampled air volume.

## 2.2 Kinetic model

Herein we used a unified vapor wall-loss transport model developed by Huang et al. (2018b) to fit the experimental

data. Figure 2 shows the concept of the model. Briefly, SVOCs partition between the gas phase and the surface of the FEP
film. Subsequently, the absorbed SVOCs may diffuse to the inner layer of the film. As the thickness of the FEP film (75 μm)
is a couple of orders larger than that of the surface layer (~ 5 nm) (Huang et al., 2018b), the inner layer is assumed as an
infinite sink. As a result, the diffusion process of SVOCs from the inner layer to the film surface is ignored. A list of all the
parameters is provided in Nomenclature. The governing equations without and with considering diffusion to the inner layer
are presented below, respectively.

(1) Without considering the diffusion process in the inner layer, the wall loss process is solely controlled by

partitioning of SVOCs between the gas phase and surface layer and can be described as follows
$$\text{Gas phase} \underset{k_{-1}}{\overset{k_1}{\rightleftharpoons}} \text{Surface} \tag{1}$$

where $k_1$ and $k_{-1}$ are forward and backward rate constants in the process. The corresponding first-order kinetic equations are
$$\frac{dC_{gas}}{dt} = -k_1 C_{gas} + k_{-1} C_{surface}$$
$$\frac{dC_{surface}}{dt} = k_1 C_{gas} - k_{-1} C_{surface} \tag{2}$$





where $C_{gas}$ and $C_{surface}$ are the SVOC concentrations in gas phase and on wall surface, respectively. It should be noted that
$C_{surface}$ was defined as the total mass of SVOC that was divided by the chamber volume, following previous studies
(Matsunaga and Ziemann, 2010; Yeh and Ziemann, 2014; Yeh and Ziemann, 2015). This model has been commonly used to
interpret the experimental data of vapor wall loss in previous studies (Matsunaga and Ziemann, 2010; Yeh and Ziemann,
2014; Yeh and Ziemann, 2015; Zhang et al., 2015).

The gas-surface equilibrium time scale $\tau_{surface}$ and equilibrium constant $K_{eq}$ can be obtained by

$$\tau_{surface} = \frac{1}{k_1 + k_{-1}} \tag{3}$$

$$K_{eq} = \frac{k_1}{k_{-1}} = \left[\frac{C_{surface}}{C_{gas}}\right]_{eq} \tag{4}$$

(2) Considering the diffusion process in the inner layer, the whole vapor wall loss process can be formulated as

follows

$$\text{Gas phase} \underset{k_{-1}}{\overset{k_1}{\rightleftharpoons}} \text{Surface} \overset{k_2}{\rightarrow} \text{Inner Layer} \tag{5}$$

where $k_2$ is the first-order loss rate constant in the diffusion process. Correspondingly, the kinetic processes for $C_{gas}$ and
$C_{surface}$ can be described by the following equations

$$\frac{dC_{gas}}{dt} = -k_1 C_{gas} + k_{-1} C_{surface}$$
$$\frac{dC_{surface}}{dt} = k_1 C_{gas} - k_{-1} C_{surface} - k_2 C_{surface} \tag{6}$$

The diffusion process has the first-order decay time scale $\tau_{inner}$ of $\tau_{inner} = \frac{1}{k_2}$. If $k_2 \ll k_1 + k_{-1}$ (i.e., $\tau_{inner} \gg \tau_{surface}$),
gas-surface partitioning occurs much faster than the diffusion process to the inner layer. In this case, the loss rate of SVOC
from the gas phase can asymptotically be represented as

$$\frac{dC_{gas}}{C_{gas}dt} \approx -\frac{K_{eq}}{1 + K_{eq}}k_2 \tag{7}$$

The data analysis and model fitting were conducted using Wolfram Mathematica 13.1. The controlling factors of

individual parameters in the above equations were previously discussed by Huang et al. (2018b).



## 3 Results and discussion

### 3.1 Wall loss of *n*-alkanes at room temperature

An example of temporal profile for $C_{14}$-$C_{19}$ *n*-alkanes during the experiment at 298 K is shown in Figure 3. The figure demonstrates the temporal change of $C_{gas}/C_0$, where $C_0$ indicates the initial concentration of *n*-alkanes. The values of $C_{gas}/C_0$ for each *n*-alkane exhibited similar patterns. During the first one hour following the injection, $C_{gas}/C_0$ exponentially decreased. After that, gradual decreases in $C_{gas}/C_0$ were observed. For example, the decline in gas fraction for $C_{14}$ *n*-alkane during the first hour accounted for 71% of the total change in $C_{gas}/C_0$ over the whole experimental period of 15 hours. The values of $C_{gas}/C_0$ decreased with the increase in carbon number, indicating enhanced wall loss. The values of $C_{gas}/C_0$ at 15 hours after injection were 0.32, 0.25, 0.16, 0.097, 0.069, and 0.037 for $C_{14}$ - $C_{19}$ *n*-alkanes, respectively.

The experimental result can be well fited using the two-layer model, but the fits deteriorate in the case that diffusion in the inner layer is neglected (Figure 3). The optimized parameter sets are shown in Table S3. Mass fractions of injected chemical species in the gas, surface, and inner layer phases that were estimated using the model are shown in Figure S4. In the case of the most volatile compound ($C_{14}$ *n*-alkane), the maximum mass fraction in the surface phase occurred at 2 hours after injection. Subsequently, the mass fractions for the compound in both gas phase and surface layer gradually decreased. During this period, the ratio of the mass in the surface layer to that in the gas phase stabilized at 1.33. The mass fraction of the compound in the inner layer steadily increased, reaching 0.22 at 15 hours after injection.

In the case of the least volatile compound ($C_{19}$ *n*-alkane), the mass fraction in the surface layer reached the maximum (~76%) approximately 1 hour after injection, accounting for the rapid decrease in the observed concentration in the gas phase. Subsequently, mass fractions of the compound in the gas phase and in the surface layer decreased in proportion, maintaining a constant ratio of the two (Figure S4). The mass fraction of the compound in the inner layer kept increasing during the experiment. At 15 hours after injection, 87% of the compound existed in the inner layer.

The time scale for *n*-alkanes to reach partitioning equilibrium between the gas and surface phases is estimated to be 12 ~ 35 mins, consistent with literature data. For example, Matsunaga and Ziemann (2010) reported that the corresponding time scale for $C_8$ - $C_{16}$ alkanes was $60 \pm 20$ mins. The corresponding value for oxygenated organic compounds was reported as $26 \pm 23$ mins (Yeh and Ziemann, 2015).

Our result for the mass transfer of SVOCs to the inner layer can also be compared with a previous study. The rates for the decrease in $C_{gas}/C_0$ for $C_{14}$-$C_{19}$ *n*-alkanes were 0.6–1.3% hour$^{-1}$ after the partitioning between gas phase and surface layer reached equilibrium (*i.e.*, 3 ~ 15 hours). Yeh and Ziemann (2015) reported the corresponding value for 2-ketones as approximately 1% hour$^{-1}$ for the time scale of 7 hours. They suggested that the value is close to the theoretical value for the Fickian diffusion loss rate (~0.5 % hour$^{-1}$).



## 3.2 Temperature dependence of wall loss of *n*-alkanes

Figure 4a summarizes the values of $C_{gas}/C_0$ for all experiments at 3 hours after injection. The data for this sampling time was selected, as the loss of gas phase species by partitioning to the surface layer accounted for the dominant portion of the decline in the gas phase concentration. It should be noted that fitting the experimental data using the two-layer model was challenging for the low-temperature experiments, as the chamber was cooled after the injection of *n*-alkanes. Potential uncertainties associated with the employment of the data at 3 hours after injection as a proxy for gas-surface partitioning are summarized in Text S1.

Generally, $C_{gas}/C_0$ was lower for less volatile compounds and at lower temperature, suggesting enhanced partitioning of *n*-alkanes to the chamber wall. The data for the room temperature ($C_{gas}/C_0$ = 0.47, 0.45, 0.34, 0.24, 0.17, and 0.091 for $C_{14}$, $C_{15}$, $C_{16}$, $C_{17}$, $C_{18}$, and $C_{19}$ *n*-alkanes) were smaller than that have been reported by a previous study. Namely, Matsunaga and Ziemann (2010) quantified the corresponding values for equilibration between the gas and surface phases for $C_{14}$-$C_{16}$ *n*-alkanes as ~80 – 90%. The enhanced partitioning to the surface layer in our study is likely due to that the chamber we used is smaller (1 m³ versus 5.9 m³).

Figure 4b shows the values of $C_{gas}/C_0$ as a function of temperature at 15 hours after injection. In all experiments, the values of $C_{gas}/C_0$ at 15 hours after injection were consistently lower than those for 3 hours. For instance, $C_{gas}/C_0$ for $C_{14}$ *n*-alkane at 262 K decreased from 0.15 (3 hours) to 0.06 (15 hours). As discussed in the case of the experiment at 298 K, the result suggests that diffusional loss in the inner layer of the chamber wall occurred for the whole temperature range.

## 3.3 Temperature dependence of partitioning between gas phase and wall surface

The temperature dependence in the data summarized in Figure 4a can be understood by considering changes in partitioning between the gas phase and surface layer. Matsunaga and Ziemann (2010) introduced the following equation for relating $C_{surface}/C_{gas}$ and temperature based on the equilibrium dissolution model:

$$\left[\frac{C_{surface}}{C_{gas}}\right]_{eq} = K_{eq} = \frac{C_{FEP\_surface}RT}{M_{wall}\gamma_{FEP\_surface}P_s(T)} \tag{8}$$

where $C_{FEP\_surface}$ is the equivalent organic mass concentration of the FEP chamber surface wall, $M_{wall}$ is the average molecular mass of the FEP, $\gamma_{FEP\_surface}$ is the activity coefficient of the organic compound in the Teflon surface, $R$ is the gas constant, and $T$ is temperature. $P_s(T)$ is the saturation vapor pressure of the compound at temperature $T$. To use Equation (8) $P_s(T)$ was calculated by the EVAPORATION group contribution method (Compernolle et al., 2011). Comparison between the EVAPORATION method with other approaches for estimating $P_s(T)$ is available in Figure S5. The value of





$[C_{surface}/C_{gas}]_{eq}$ was approximated using $1/[C_{gas}/C_0]_{\text{at 3 hours}} - 1$ by assuming that diffusion of $n$-alkanes to the inner layer
was still a minor loss process within 3 hours. Among the terms for the right-hand-side of equation (8), $RT/P_s(T)$ can be
calculated from the experimental conditions. The equation suggests that $[C_{surface}/C_{gas}]_{eq}$ and $RT/P_s(T)$ may linearly
correlate with the slope of $C_{FEP\_surface}/(M_{wall}\gamma_{FEP\_surface})$.
Figure 5 shows the correlations between $C_{surface}/C_{gas}$ and $RT/P_s(T)$ for individual compounds. For all the tested
compounds, these two parameters correlated well, even though $C_{surface}/C_{gas}$ increased by more than one order of magnitude
when the chamber was cooled down. The result suggests that equation (8) can be applied to a wide range of temperatures
without considering the temperature dependence of $C_{FEP\_surface}/(M_{wall}\gamma_{FEP\_surface})$ to account for partitioning of a chemical
species to the surface layer. In other word, $\gamma_{FEP\_surface}$ can be practically treated as a constant for the investigated temperature
range, given $C_{FEP\_surface}$ and $M_{wall}$ are independent of temperature. This implication is consistent with previous findings that
the activity coefficients of organic compounds in polymers only change slightly with temperature. For instance, Kontogeorgis
et al. (1993) compared the experimental and modelled values of activity coefficients for hydrocarbons in a few polymers such
as low-density polyethylene. The values of activity coefficients change by 10~20% for a temperature change of 100 K.
Values of $\gamma_{FEP\_surface}$ for $n$-alkanes can be estimated from Figure 5. Based on equation (8), the fitted slopes
correspond to $C_{FEP\_surface}/(M_{wall}\gamma_{FEP\_surface})$. For a specific chamber design, compound-independent $C_{FEP\_surface}$ can be
estimated by the density of FEP film (2150 kg m$^{-3}$) and the thickness of surface layer ($\sim$ 5 nm) (Huang et al., 2018b). For the
chamber in this experiment, $C_{FEP\_surface}$ = 78.2 mg m$^{-3}$. For estimating compound-dependent $\gamma_{FEP\_surface}$, previous studies
assumed $M_{wall}$ = 200 g mol$^{-1}$ (Huang et al., 2018b; Matsunaga and Ziemann, 2010). The same approximations were employed
in the present study.
Figure 6 plots the retrieved values of $\gamma_{FEP\_surface}$ for $n$-alkanes against $P_s(298 \text{ K})$ for $n$-alkanes. The figure also
shows the corresponding parameters obtained from previous experimental studies (Matsunaga and Ziemann, 2010; Yeh and
Ziemann, 2014; Yeh and Ziemann, 2015; Krechmer et al., 2016). Regardless of differences in types of chemicals and chambers,
the experimentally estimated values of $\gamma_{FEP\_surface}$ and $P_s(298 \text{ K})$ correlate in logarithmic axes. The relationship followed
the equation of $\ln(\gamma_{FEP_{surface}}) = 0.40 - 0.61\ln(P_s(298 \text{ K}))$.
**3.4 Characterization of diffusion from the Teflon surface to inner layer**
Values of $k_2$ were estimated using equation (7), since values of $\tau_{inner}$ are at least 18 times larger than those of
$\tau_{surface}$ (Table S3). The values of $C_{gas}/C_0$ at 3 hours after injection were employed to calculate $K_{eq}$ as discussed earlier.
The experimental data for 9, 12, and 15 hours after injection was employed for obtaining $k_2$.



Figure 7 plots the estimated values of $k_2$ against $P_s(T)$ for all compounds in all experiments. The values of $k_2$ and
$P_s(T)$ positively correlate. As a comparison point, a previous study reported positive correlations for (1) the diffusivity of
organic compounds in FEP film and saturation concentration, and (2) $k_2$ and diffusivity (Huang et al., 2018b). Our current
result is qualitatively similar to the previous study, though temperature was maintained as a constant in the previous study.
The decrease in $k_2$ at lower temperature could be induced by reduced viscosity in the inner layer or weakened thermal motion
of $n$-alkane molecules. Further research, that incorporates changes in FEP film properties with temperature would be needed
in the future for quantitatively interpreting the data.
**4 Conclusions**
The present study investigated the wall loss process of $C_{14}$-$C_{19}$ $n$-alkanes to the wall of a 1 m$^3$ chamber bag, which was
composed of the FEP film. The temperature of the chamber was controlled for the range of 262 to 298 K. Decay in gas-phase
concentrations of the $n$-alkanes was quantified using the SV-TAG for 15 hours following injection. The temporal variations in
the $n$-alkane concentrations suggested two types of loss processes. The first process was characterized by rapid exponential
decay in the first few hours. Subsequently, slow first-order decreases in the $n$-alkane concentrations were identified until the
end of the experiment. Enhanced wall loss was observed at lower temperatures for all compounds.
The experimental data were well fitted using the two-layer kinetic model, which considers partitioning of gas-phase
species to the surface layer of the chamber film and further diffusion to the inner layer. The analysis suggests that when the
Teflon bag chamber is operated at low temperatures, partitioning of gas phase species to the chamber wall surface is enhanced,
whereas the permeation of the chemical compounds to the inner layer is suppressed. The temperature effect on gas-surface
partitioning overweighs that on diffusion into the inner layer for $n$-alkanes, leading to an overall enhanced wall loss at lower
temperature.
The quasi-equilibrium partitioning of $n$-alkanes between the gas phase and surface layer was interpreted by considering
the dissolution process of the species into the surface layer. Values of $C_{surface}/C_{gas}$ at quasi-equilibrium are proportional to
$RT/P_s(T)$ for individual compounds. The result suggests that decreased saturation vapor pressure is the major driving force
for enhanced partitioning to the surface layer at low temperatures for all investigated compounds, while their activity
coefficients can be practically treated as constants for the investigated temperature range. The relationship can be potentially
employed for predicting changes in wall loss of SVOCs as a function of temperature, after further verification employing other
types of organic compounds.
In the future, the underlying mechanisms of the present findings will need to be sought for a better understanding of the
chamber wall loss of SVOCs. The present study focused on $n$-alkanes. In the case of chamber experiments for SOA formation,
wall loss processes of oxygenated chemical species would be more important. Thus, a temperature-dependent wall loss study



for oxygenated chemical species will still need to be conducted for interpreting SOA chamber experiments under a wide range
of temperatures.

**Data Availability**

Data will be made available on request.

**Author contribution**

**Longkun He:** Conceptualization, Methodology, Experiment, Data curation, Formal analysis, Writing – original draft. **Wenli**
**Liu:** Methodology, Experiment, Writing – review & editing. **Yatai Li:** Methodology, Writing – review & editing. **Jixuan**
**Wang:** Experiment, Writing – review & editing. **Mikinori Kuwata:** Conceptualization, Methodology, Project administration,
Funding acquisition, Formal analysis, Writing – review & editing, Supervision. **Yingjun Liu:** Conceptualization,
Methodology, Project administration, Funding acquisition, Formal analysis, Writing – review & editing, Supervision.

**Competing interests**

The authors declare that they have no conflict of interest.

**Acknowledgements**

We thank Dr. Ying Zhou for assisting to improve figure quality. This research was supported by the National Natural
Science Foundation of China (92044303, 42175121, and 42150610485).



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





**Nomenclature**
A table that contains the definitions of parameters and corresponding units.
$k_1$            forward rate constant (min$^{-1}$)
$k_{-1}$           backward rate constant (min$^{-1}$)
$k_2$            first-order loss rate constant (min$^{-1}$)
$\tau_{surface}$       gas-surface equilibrium time scale (min)
$\tau_{inner}$        diffusion time scale (min)
$C_0$            initial SVOC concentration in gas phase (μg m$^{-3}$)
$C_{gas}$          SVOC concentration in gas phase (μg m$^{-3}$)
$C_{wall}$         SVOC concentration on wall surface (μg m$^{-3}$)
$K_{eq}$           gas-surface equilibrium constant
$C_{FEP\_surface}$     equivalent organic mass concentration of the FEP chamber surface (mg m$^{-3}$)
$M_{wall}$         average molecular mass of the Teflon wall (g mol$^{-1}$)
$\gamma_{FEP\_surface}$     activity coefficient in the Teflon surface
$R$             gas constant (J K$^{-1}$ mol$^{-1}$)
$T$             temperature (K)
$P_s(T)$         saturation vapor pressure of compound at temperature $T$ (Pa)





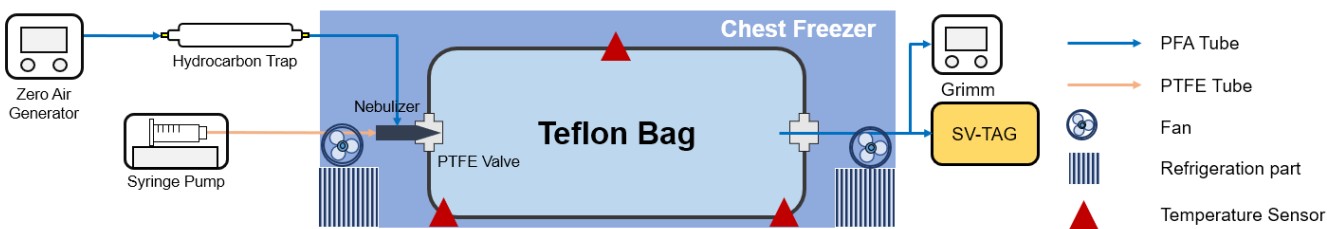


**Figure 1.** Schematic diagram of the experimental setup.





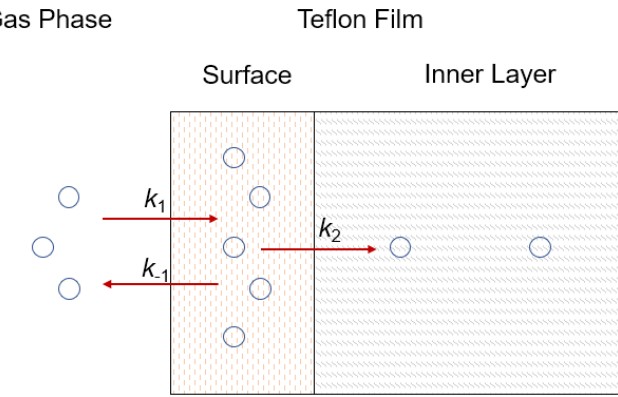


**Figure 2.** Schematic diagram of wall loss process. Compounds partition between gas phase and surface layer with forward and backward rates ($k_1$ and $k_{-1}$). Compounds in surface layer undergo irreversible diffusion into inner layer with first-order loss rate ($k_2$).





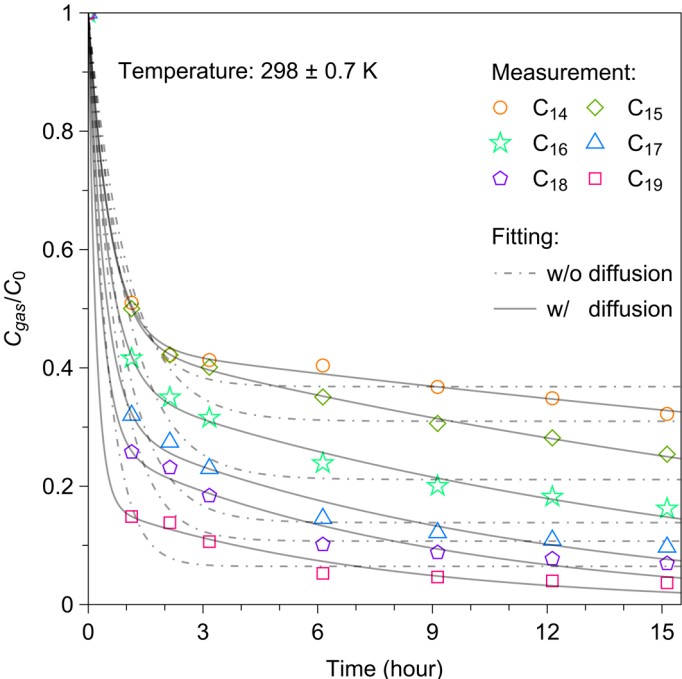

**Figure 3.** Temporal variation in $C_{gas}/C_0$ for C$_{14}$-C$_{19}$ $n$-alkanes at 298 ± 0.7 K following injection. $C_{gas}$ is the concentration of each $n$-alkane in the gas phase, and $C_0$ is the corresponding initial concentration of each $n$-alkane. The two-layer kinetic sorption model (Section 2.2) was employed to fit the data (black solid line). The black dot-dashed lines show the fitting result to the model that ignores the diffusion process to the inner layer (*i.e.*, $k_2 = 0$).





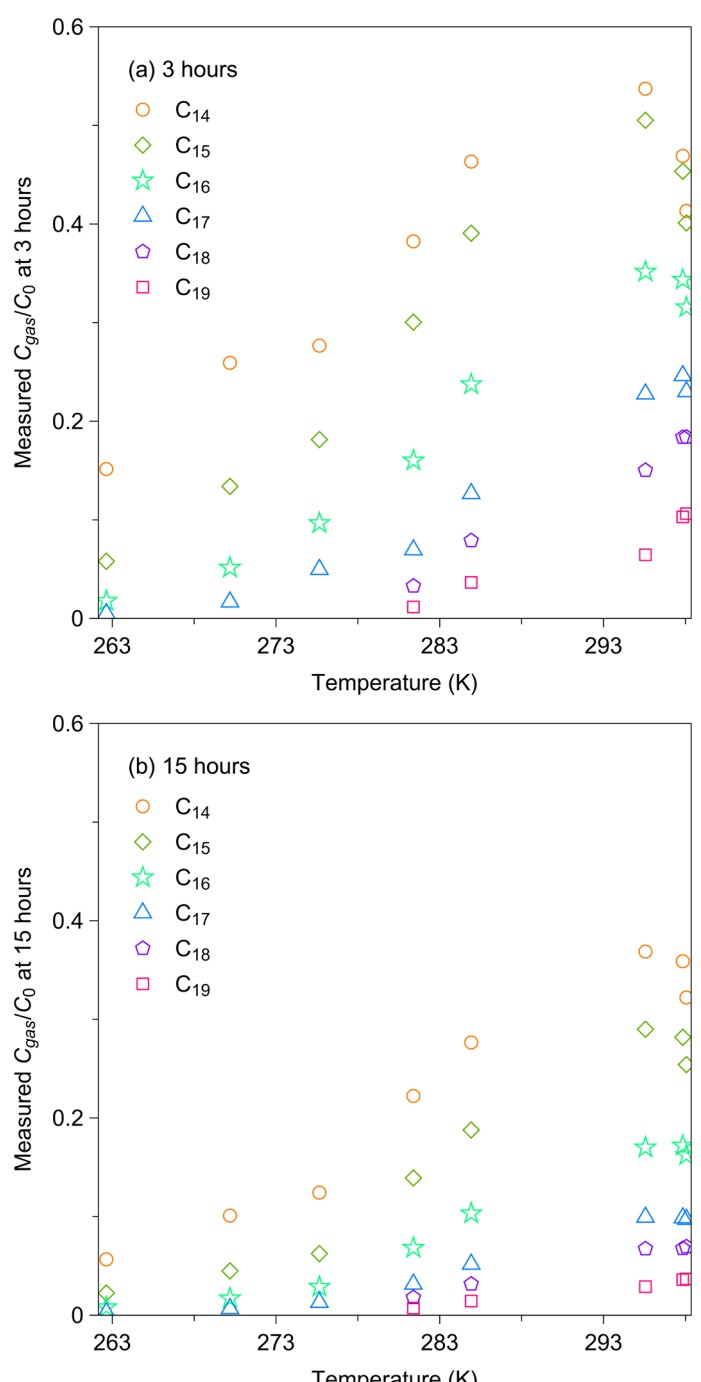

403

**Figure 4.** Measured values of $C_{gas}/C_0$ at (a) 3 hours and (b) 15 hours after injection.

404

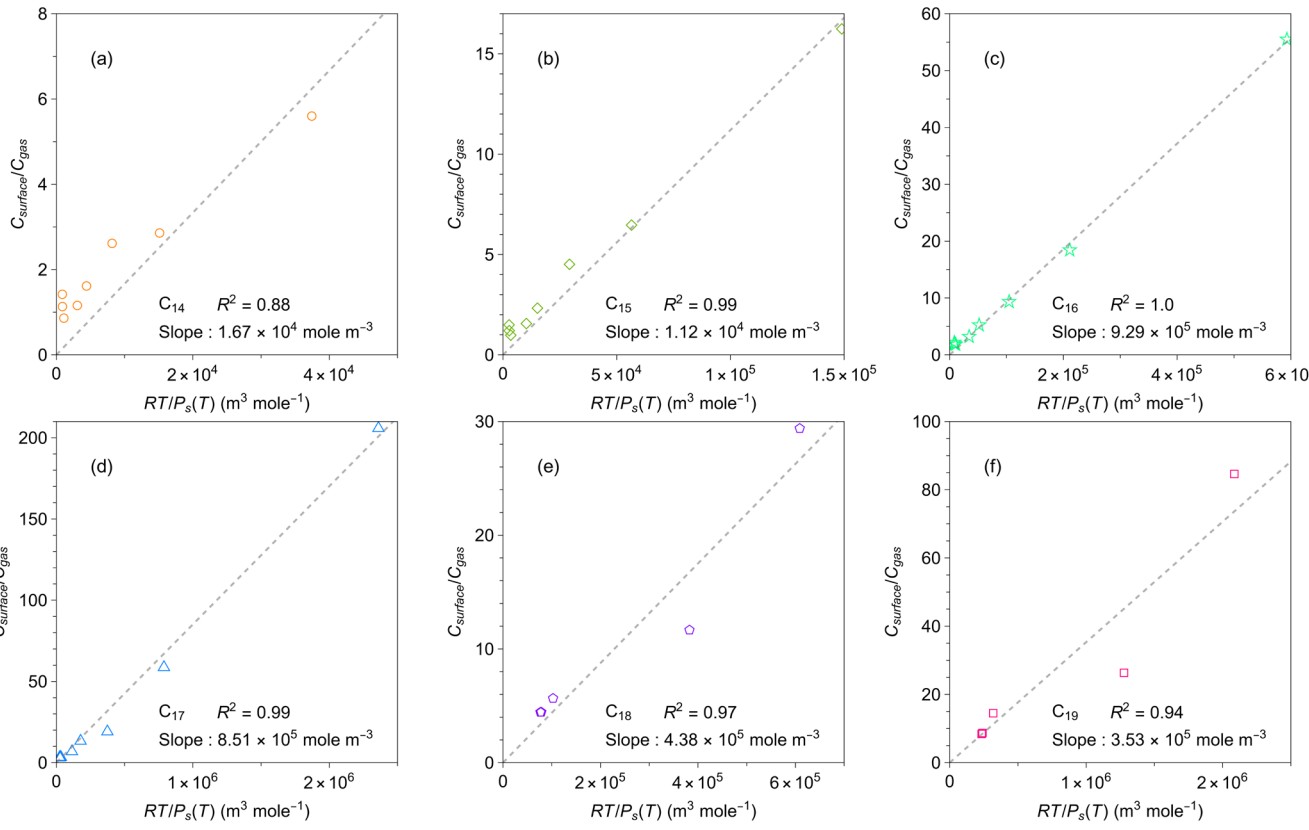

405

**Figure 5.** Relationships between measured ratio of concentrations in the chamber wall surface phase and in the gas phase at quasi-equilibrium and calculated values of $RT/P_s(T)$ for individual $n$-alkanes. Calculation methods for $C_{surface}/C_{gas}$ is detailed in the text. The values of $RT/P_s(T)$ for each $n$-alkane were calculated by the EVAPORATION group contribution method (Compernolle et al., 2011). The black dashed lines are linear least-squares that fit the data for each $n$-alkane.





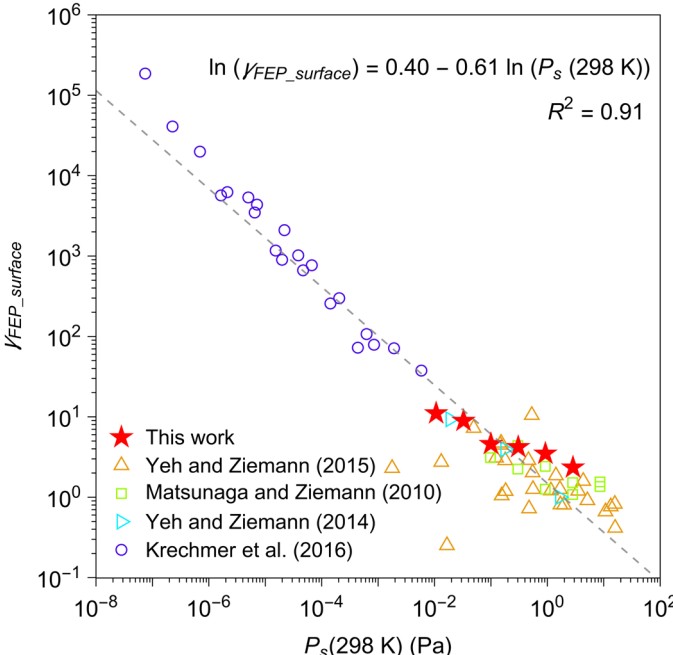

**Figure 6.** Activity coefficient ($\gamma_{FEP\_surface}$) of organic compounds in FEP film. The sources of data include this work and the literature (Matsunaga and Ziemann, 2010; Yeh and Ziemann, 2014; Yeh and Ziemann, 2015; Krechmer et al., 2016). A list of chemical species that were investigated by each study is available in Table S4. Saturation vapor pressures at 298 K ($P_s$ (298 K)) were estimated by EVAPORATION (Compernolle et al., 2011).



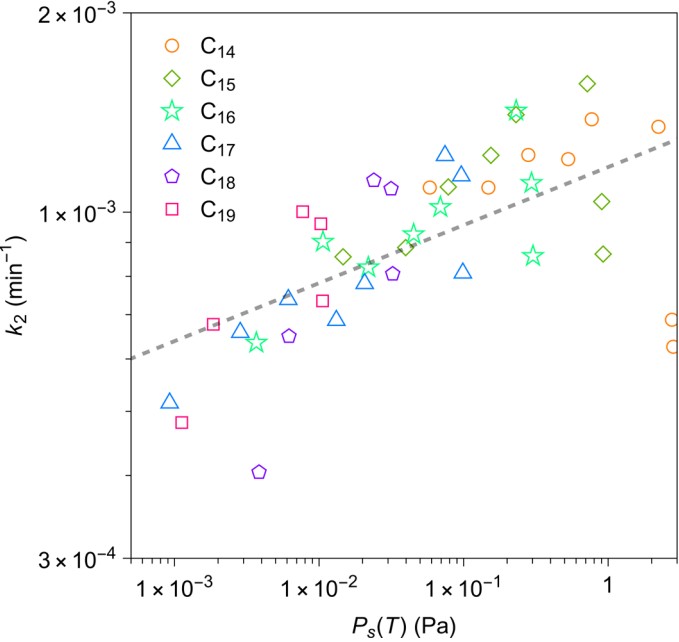

415

**Figure 7.** Relationship between calculated first-order loss rate $k_2$ for each *n*-alkane and calculated values of saturation vapor pressure by the EVAPORATION group contribution method (Compernolle et al., 2011). The calculation method for $k_2$ is detailed in the text. The black dashed line is a linear least-squares fit to the data in a logarithmic scale.