# Peer review of "Wall loss of semi-volatile organic compounds in a Teflon bag chamber"

_Atmospheric Measurement Techniques, 2023_

## Author Comment (AC1)

We appreciate the reviewer for providing us useful comments. In the following, original reviewer comments, author's responses, and corresponding updates on the main text are shown as purple, black, and *italic*. Line numbers in the responses correspond to those in the originally submitted version.

**Reviewer #1**

General Comments

He et al. provide a good preprint in terms of scientific significance and presentation quality. However, I feel that detail is lacking, making scientific quality fair. I cannot currently recommend publication in Atmospheric Measurement Techniques until the specific points below are addressed, some of which I consider to be major revisions, though the editor may consider as minor.

We thank the reviewer for the positive comments on our scientific significance and presentation quality. Responses to the individual comments are given below.

Specific Comments

(1.1) He et al. verify that a two layer model, as detailed in Huang et al. (2018) (https://doi.org/10.1021/acs.est.7b05575), with parameter fitting, can reproduce chamber observations of gas-wall partitioning under varying temperatures. Furthermore, they demonstrate that the absorption and diffusion mechanisms of wall loss of uncharged organic molecules have opposite responses to temperature change under dry conditions. The finding around diffusion response to temperature is novel, and it is in the interests of the chamber community that it is published.

Their observations that wall loss of semi-volatile organics is enhanced at lower temperatures is not, to my knowledge, novel, and has been reported by Zhang et al. (2015) (https://doi.org/10.5194/acp-15-4197-2015) (who identify a link with pure component saturation vapour pressure, but don't conclude a direct causal relation with pure component saturation vapour pressure), and is observed in the supporting material of Huang et al. (2018) (https://doi.org/10.1021/acs.est.7b05575) (though they did not attribute the cause to changed pure component saturation vapour pressure). Furthermore, Matsunaga and Ziemann (2010) (https://doi.org/10.1080/02786826.2010.501044) show the relationship between fraction partitioned to wall and component saturation vapour pressure in their Figure 6, implying that whether the saturation vapour pressure changes because of a change of component or because of a change in temperature, the partitioning will change accordingly.

However, Huang et al. (2018) (https://doi.org/10.1021/acs.est.7b05575) state that the effect of temperature on gas-wall partitioning requires further study, and this paper is the first to my knowledge to fulfil this request and provide mechanistic insight. It is therefore a valuable paper.

We thank the reviewer for the positive comments. As the reviewer pointed out, Zhang et al. (2015) conducted their experiments for wall loss at 298 K and 318 K. On the other hand, we decreased the temperature for the tropospheric relevant range. Our intention for the present study was to investigate wall-loss process as a function of temperature, as Huang et al. (2018) suggested. The following sentences were added to the revised for stressing the point:

"*This study investigated vapor wall loss of $C_{14}$-$C_{19}$ n-alkanes in a Teflon chamber for the temperature range of 262*

*to 298 K by monitoring the evolution of their gas-phase concentrations following a pulse release. The wall-loss process was investigated as a function of temperature. The experimental results were analyzed using the two-layer kinetic model, which considers partitioning of gas phase SVOCs to the surface layer, as well as further diffusion to the inner layer. Temperature effects on the two processes were evaluated separately.*" (Lines 56-59)

(1.2) I therefore recommend that the title be changed to indicate that mechanistic insight is presented. This way readers will be directed to this article when interested in the mechanisms of gas-wall partitioning in Teflon chambers.

We thank the reviewer for this suggestion. We concur that it is better to clearly indicate the presentation of mechanistic insight in the title. As per the recommendation, we have revised the title as below:

"*Wall loss of semi-volatile organic compounds in a Teflon bag chamber for the temperature range of 262-298 K: mechanistic insight on temperature dependence*" (Lines 1-2)

(1.3) I recommend that units in terms of minutes be converted to seconds (e.g. k1, k-1, k2), to be consistent with the literature (e.g. Huang et al. (2018) (https://doi.org/10.1021/acs.est.7b05575)).

We thank the reviewer for this input. We have converted all units in terms of minutes to be seconds.

(1.4) I also recommend that the mechanistic aspect of greatest novelty – the diffusion through the Teflon interior – is expanded on in the paper. It seems quite feasible to plot k2 as a function of the effective diffusion coefficient of organics through the Teflon, along with a line/curve of best fit and its coefficients, as Huang et al. (2018) (https://doi.org/10.1021/acs.est.7b05575) do in their Figure 5. Additionally, then plotting diffusion coefficient (rather than k2 as currently given in Figure 7) against component saturation vapour pressure allows for a better mechanistic understanding and much easier comparison with other publications. The text of the 'Results and discussion' section should then be updated to reflect these new figures. If these changes around diffusivity cannot be implemented, the paper should explain why so that future research is able to improve on these experiments. In addition, if these changes around diffusivity cannot be implemented, then the abstract should be changed to discuss the Teflon inner layer interaction in broader terms, rather than inferring that diffusion is the key process in the interaction.

We thank the reviewer for this suggestion. Presenting diffusion coefficient is indeed a valuable suggestion. In our experimental protocol, however, cooling of the chamber was started 1 hour later after injection. Chamber temperature decreased from room temperature to set temperature, which meant that diffusion coefficient would change correspondingly. It is therefore difficult to estimate diffusion coefficients. We explained the reason in the main text and changed the abstract to discuss the Teflon inner layer interaction in broader terms. The text is revised as below:

"*On the contrary, diffusion process of n-alkanes from the surface to inner layer slowed down at reduced temperature. Hence the relative importance of the surface and inner layers on wall-loss process changes with temperature. Mechanistic studies on these processes will need to be conducted in the future to quantitatively predict the influence of temperature-dependent wall-loss processes of SVOCs on laboratory experimental results.*" (Lines 23-26)

"*It should be noted that fitting the experimental data using the two-layer model was challenging for the low-temperature experiments, as the chamber was cooled after the injection of n-alkanes. Since the chamber was cooled after the injection of n-alkanes, the model parameters would change correspondingly with chamber temperature.*"

(Lines 167-168)

(1.5) He et al. make no mention of the effect of ageing (in their case previous experiments involving injection of alkanes), but for a paper discussing wall losses, this should be at least discussed, if not quantified, as it is in Matsunaga and Ziemann (2010) and in Huang et al. (2018).

We thank the reviewer for this input. Our chamber was newly purchased for the experiment, meaning that it was employed for no other experiments. As for the cases you mentioned, after each experiment, the chamber was heated to ~320 K and continuously flushed by purified air. The cleaning process would last for 2-3 days until the gas concentration of target alkanes decreased to the background level. We added the information of ageing in the text:

"*Prior to each experiment, the chamber was heated to ~320 K and continuously flushed using purified air;. The cleaning process lasted for 2~3 days until the concentration of investigated n-alkanes dropped to the background level.*" (Lines 77-79)

(1.6) The area/volume ratio of a chamber is frequently described in other papers as a key determinant in gas-wall partitioning, therefore to aid readers in their interpretation, it would be useful to have this value given in units of /m in the section describing the chamber. I also find the explanation of enhanced partitioning compared to the Matsunaga results on line 175 to be too brief, and wonder whether the authors could either explain why chamber volume makes a difference, or consider an explanation in terms of area/volume rather than just volume.

We thank the reviewer for this input. We added the area/volume ratio in the Method section and in the comparison with previous results to better explain the enhanced partitioning. The text is revised as below:

"*Figure 1 shows the experimental setup. The experiment was conducted using a fluorinated ethylene propylene (FEP) bag with the volume of 1 $m^3$. The thickness of the FEP film for the bag was 75 μm. The dimension of the bag was 260 cm $\times$ 55 cm $\times$ 70 cm. The area to volume ratio of the chamber was 7.26 $m^{-1}$.*" (Lines 62-64)

"*The enhanced partitioning to the surface layer in our study is likely due to that the chamber for the present study we used is smaller (1 $m^3$ versus 5.9 $m^3$) has a larger area to volume ratio (7.26 $m^{-1}$ versus 3.39 $m^{-1}$).*" (Lines 175-176)

(1.7) Line 88 says that air leaked out of the bag. The authors must provide at least qualitative, and preferably quantitative, evidence that this did not significantly affect the concentration of alkanes in the bag, otherwise a leak of alkanes from the bag could cause the observed decreases in concentration with time, rather than gas-wall partitioning. On this point, have the authors considered that the removal of air from the bag for instrument sampling led to decreased pressure in the bag (rather than air leaks), which in turn led to compression of the bag volume? In this case, pressure inside the bag may have been maintained and therefore gas-phase concentration of alkanes were unaffected by changes in bag volume.

We thank the reviewer for this input. Compression of bag volume was observed during our experiments. But the pressure inside the bag would be maintained. Additionally, our measurements for phthalates ensured room air would not penetrate into the chamber. The gas-phase concentrations of alkanes were therefore unaffected by the changes in bag volume. However, compression of bag volume would lead to an increase in the area to volume ratio. Consequently, this could disrupt the gas-wall interaction, especially the relatively slow diffusion process. This could lead to an

misestimation of diffusion loss rate. We acknowledged that increase of area to volume ratio was a quite challenge. Future studies may benefit from addressing this issue. Based on some photos during the experiment, the leak-out of air could have increased the area to volume ratio by a factor of few. The point was clarified in the revised manuscript:

"*Although the air in the bag leaked out during experiments due to compression of the bag by its own weight, absence of intrusion of room air to the bag was confirmed by observing no changes in contaminant signals (Table S2). The gas-phase concentrations of n-alkanes were therefore unaffected by the changes in bag volume.*" (Lines 88-89)

"*The decrease in $k_2$ at lower temperature could be induced by reduced viscosity in the inner layer or weakened thermal motion of n-alkane molecules. It should be noted that compression of bag volume during experiment would lead to an increase in the area to volume ratio. Consequently, this could disrupt the relatively slow diffusion process. Based on some photos during the experiment, the leak-out air could have increased the area to volume ratio by a few factors. Further research, that incorporates changes in FEP film properties with temperature would be needed in the future for quantitatively interpreting the data. Further study, that incorporates considering changes of chamber volume, would be needed in the future for quantitatively interpreting the data.*" (Lines 223-225)

(1.8) Line 66 says fans were used, and Figure 1 shows these fans to be outside the bag but inside the chamber housing. The authors must mention how the resulting buffeting of the Teflon bag affects the rate of mixing of air in the bag and therefore the rate of gas-wall partitioning, especially when comparing to other results, such as Matsunaga and Ziemann (2010).

We thank the reviewer for pointing out this question. The fans were used to mixing the air outside the chamber, in order to make air temperature in the freezer to be uniform. As for the mixing of air in the bag, our $CO_2$ experiment demonstrated that it took ~30 mins for $CO_2$ to be well mixed in the bag (Fig. S1). No fan was installed inside of the Teflon bag. To clarify, we revised the manuscript:

"*Two fans were installed in the freezer (outside the bag) to promote the mixing of the air so that air temperature in the freezer was uniform.*" (Lines 66-67)

"*The chamber volume was experimentally validated by employing CO₂ as a tracer (Figure S1). The timescale for CO₂ to be well mixed in the bag after a pulse injection was approximately 30 mins (Figure S1).*" (Line 64)

(1.9) Given that some of these recommendations are substantial, I expect that the authors will consider making appropriate changes to text throughout the paper beyond the specific locations of text mentioned here.

We thank the reviewer for the valuable inputs. We fully revised the manuscript based on the comments.

Technical Corrections

(1.10) Line 12: consider 'The wall-loss process' rather than 'Wall-loss process'

We thank the reviewer for the suggestion. The text is revised as below:

"*The wWall-loss process of gas-phase species in Teflon bag chambers has typically been investigated at around room temperature.*" (Line 12)

(1.11) Line 23: consider 'diffusion of n-alkanes' rather than 'diffusion process of n-alkanes'

We thank the reviewer for the suggestion. The text is revised as below:

"*On the contrary, diffusion process of n-alkanes from the surface to inner layer slowed down at reduced temperature.*" (Line 23)

(1.12) Equation 7 (Line 131): should the Cgas term be present in the divisor on the left hand side?

We thank the reviewer for the suggestion. Left side of Equation 7 represented the apparent first-order decay loss constant (Huang et al. 2018). To clarify, we revised the description as below:

"*In this case, the apparent first-order decay loss constant the loss rate of SVOC from the gas phase can asymptotically be represented as (Huang et al., 2018b):*" (Lines 129-130)

(1.13) Line 143: 'fitted' rather than 'fited'

We thank the reviewer for the suggestion. The text is revised as below:

"*The experimental result can be well fitted using the two-layer model,*" (Line 143)

(1.14) Line 228: consider 'composed of FEP film' rather than 'composed of the FEP film'

We thank the reviewer for the suggestion. The text is revised as below:

"*The present study investigated the wall loss process of $C_{14}$-$C_{19}$ n-alkanes to the wall of a 1 $m^3$ chamber bag, which was composed of the FEP film.*" (Line 228)

---

## Author Comment (AC2)

We appreciate the reviewer for providing us useful comments. In the following, original reviewer comments, author's responses, and corresponding updates on the main text are shown as purple, black, and *italic*. Line numbers in the responses correspond to those in the originally submitted version.

**Reviewer #2**

(2.1) This manuscript investigates the wall loss of organic gases in FEP Teflon chambers at temperatures below 298 K. This is a previously identified gap in our understanding of vapor wall loss in chamber simulations of atmospheric chemistry, and the importance of this question makes studies such as this one essential. The manuscript provides good, useful information about the effects of lower temperature on wall loss, but the study design leaves some significant gaps in understanding that could be improved with revision and potentially some additional experiments.

We thank the reviewer for providing us insightful comments. Responses to the individual comments are given below.

(2.2) Critically, the authors first inject n-alkanes at room temperature and then spend several hours cooling the chamber. Given prior literature on the effect of partitioning to Teflon polymer at temperatures above 298 K, our knowledge of how diffusion coefficients of organic compounds in polymers vary with temperature, and the other conclusions drawn by the authors from this data, I am concerned that the partitioning observed at t = 3 hours may be highly path-dependent, and might differ from the partitioning that would occur if the alkanes had been injected into a pre-cooled chamber.

We thank the reviewer for the insightful comment. We compared the results in pre-cooled chamber (at 270 K) with the post-cooled experiment (at 270.2 K) which chamber was cooled down 1 hour after injection. The comparison was presented in the revised manuscript and Figure S2. Generally, the experimental results for pre- and post- cooled experiment provided similar outputs. Results in post-cooled experiments (solid points in Figure S2) align well with the two-layer kinetic model fitting results of pre-cooled experiment (black lines in Figure S2). Specifically, the values of $C_{gas}/C_0$ at 206 min (measurement timepoint in post-cooled experiment) were 0.26, 0.14, 0.05, and 0.02. While the corresponding values calculated by the fitted model for pre-cooled experiment were 0.20, 0.13, 0.06, and 0.03. The consistency between pre- and post- cooled experiment demonstrated the validity of employing post-cooled operation procedure. It should be noted that in the pre-cooled chamber, particles were detected by OPC to be 0.9 µg m$^{-3}$. As a precaution, in the following experiments, cooling was conducted 1 hour after injection to avoid particle formation. We added this discussion in the main text and Figure S2 in supporting information:

In main text:

"*For experiments below room temperature, the cooling system of the freezer was turned on one hour after the completion of the injection. The operation procedure was employed to avoid homogeneous nucleation and subsequent condensational growth of aerosol particles. The validity of employing this post-cooled operation procedure was demonstrated by comparison with a pre-cooled chamber result at 270 K (Figure S2).*" (Lines 83-85)

Added in supporting information:

[Figure]

Figure S2. Comparison of $C_{gas}/C_0$ between pre-cooled experiment at 270 K (hollow points) and post-cooled experiment at 270.2 K (solid points). The two-layer kinetic sorption model was employed to fit the pre-cooled chamber data (black solid lines). Results in post-cooled experiments (solid points) aligns well with the two-layer kinetic model fitting results of pre-cooled experiment (black lines). The consistency between pre- and post- cooled experiment demonstrated the validity of employing post-cooled operation procedure.

(2.3) Investigation of gas-wall partitioning in Teflon tubing at 120 C (Mattila et al https://doi.org/10.1080/10962247.2023.2174612) has shown that partitioning delays did not vary with temperature. This indicated that the increase in vapor pressure of analytes was largely compensated by an increase in C_w (or C_FEP_Surface in the authors' notation). This result suggests that at lower temperatures, one might observe that the decrease in vapor pressure would be offset by a *decrease* in C_w. However, the authors here observe consistent slopes in Figure 5, indicating that the C_FEP_surface/gamma ratio is remaining constant with temperature. And following the authors literature review, the lack of temperature dependence in gamma suggests that C_FEP_Surface is constant below 298 K.

We Thank the reviewer for introducing us the very interesting paper. Mattila et al. (2023) investigated the sampling delay in measuring per- and polyfluoroalkyl substances (PFAS) by PFA tubing at 30 and 120 °C. The major differences from our present study include the temperature range and differences in chemical characteristics between *n*-alkanes and PFAS. Our conclusion about the constant C_FEP_surface was derived solely based on the experimental result for *n*-alkanes. We admit that further studies that employs chemical species that have different characteristics would be needed in the future for understanding the phenomenon in more detail. The following statement was provided in the revised manuscript for addressing the point:

"*The result suggests that equation (8) can be applied to a wide range of temperatures without considering the temperature dependence of $C_{FEP\_surface}/(M_{wall}\gamma_{FEP\_surface})$ to account for partitioning of  n-alkanes to the surface layer.*" (Lines 197-199)

"*The values of activity coefficients change by 10~20% for a temperature change of 100 K. Further studies, that*

*employ different chemical species, would be needed in the future for validating and applying the relation to a wide range.*" (Line 203)

(2.4) The authors' retrieval of k_2 (the first-order rate constant representing diffusion in to the bulk polymer) in Figure 7 indicates that at lower temperatures the rate of diffusion into the polymer is slowed. This is consistent with the results of Matilla et al: where at high temperatures there is more polymer available for partitioning, and then here at lower temperatures there is a restriction in the movement of analyte through the Teflon.

We thank the reviewer for the valuable input. Our results about diffusion rate constants are consistent with the results of Mattila et al. (2023). We cited Mattila et al. (2023) in our revised text for supporting the discussion:

"*The decrease in $k_2$ at lower temperature could be induced by reduced viscosity in the inner layer or weakened thermal motion of n-alkane molecules (Mattila et al., 2023).*" (Lines 223-224)

(2.5) The potential mechanism that concerns me is that C_FEP_Surface may be lower at colder temperatures (less polymer available for sorption), but since the authors' experimental design first establishes equilibrium at higher temperatures, there is 'extra' alkane locked up in the polymer at depths that would not be accessible within 3 h if the injection had been done in a pre-cooled freezer. Given the long timescales for evaporation from FEP, it seems plausible that as the chamber is cooled, the diffusion coefficients drop, and the surface layer potentially shrinks there could be significant mass transfer limitations keeping sorbed alkane at a given depth. In a two-layer model, this would be equivalent to transfer into the bulk polymer due to a shift in the dividing point between the two layers. This would significantly overstate the amount of wall loss compared to a case where the chamber was already cooled at the time an S/IVOC was introduced.

To support the authors' conclusion that C_FEP_Surface is constant with temperature, authors need to demonstrate that the amount of alkane sorbed in the walls at 3 hours is not a path-dependent process. Ideally this would be done through injection of analyte into a pre-cooled chamber. This could be done with just the most volatile alkanes to avoid any issues with nucleation.   If this is not an option, another approach would be showing that the partitioning equilibrium at 3 hours is not dependent on cooling rate. In the methods, the authors wait an hour after injecting before cooling the freezer. Alternate approaches could be eliminating this one hour wait; and conversely slowing the cooling rate by gradually lowering the setpoint of the freezer. If the authors observe that the partitioning at 3 hours is consistent across these cases, it would be strong evidence in support of their result that C_FEP_Surface does not vary with temperature. If faster cooling (or pre-cooling) gives a significant decrease in the amount of alkane sorbed, that would indicate that there is a strong temperature dependence in C_w, consistent with prior literature.

We thank the reviewer for the constructive comment and suggestion. As previously mentioned in response to comment (2.2), we have conducted the pre-cooled chamber experiment at 270 K for comparison. The experimental result was comparable to that of the post-cooled experiment, demonstrating the validity of employing the post-cooled experiment. We revised the main text and supporting information as mentioned in our responses to comments (2.2) and (2.3).

(2.6) Line 171: Prior literature has established that C_FEP_Surface does in fact scale directly with chamber surface area to volume (SAV) ratio. Authors should directly compare the SAV ratio of the Matsunaga & Ziemann chamber to their own, and report if the C_FEPSurface results are consistent.

We thank the reviewer for this input. As described in the response to comment (1.6), We added the SAV ratio in

the Method section and in the comparison with previous results to better explain the enhanced partitioning. The text is revised as below:

"*Figure 1 shows the experimental setup. The experiment was conducted using a fluorinated ethylene propylene (FEP) bag with the volume of 1 m³. The thickness of the FEP film for the bag was 75 μm. The dimension of the bag was 260 cm × 55 cm × 70 cm. The area to volume ratio of the chamber was 7.26 m⁻¹.*" (Lines 62-64)

"*The enhanced partitioning to the surface layer in our study is likely due to that the chamber for the present study we used is smaller (1 m³ versus 5.9 m³) has a larger area to volume ratio (7.26 m⁻¹ versus 3.39 m⁻¹).*" (Lines 175-176)

(2.7) Line 207: How does the C_FEPSurface value compare to the Huang et al recommendation of C_w = 10.8 * A / V ?

We thank the reviewer for pointing out this question. We assumed thickness of the surface layer (~5 nm) and the density of FEP film (2150 kg m⁻³), following Huang et al. 2018. Therefore, our C_FEP_Surface value was the same as recommendation of C_w. To clarify, we revised the text as below:

"*For the chamber in this experiment, $C_{FEP\_surface}$ =was assumed as 78.2 mg m⁻³, following the recommendation by Huang et al. (2018b).*" (Line 207)

(2.8) Line 211: Literature values for gamma_inf are all calculated within Huang et al. 2018, who assume a fixed C_FEP_Surface (C_w). References and phrasing here should be updated to reflect the source of the calculated gamma_inf values and also mention the assumed C_w value.

We thank the reviewer for this suggestion. We cited Huang et al. 2018 and revised the description in manuscript to reflect the source of the calculated gamma_inf values and to mention the assumed C_w value. The text is revised as below:

"*The figure also shows the corresponding parameters obtained from previous experimental studies (compiled by Huang et al. (2018b), including Matsunaga and Ziemann (2010), Yeh and Ziemann (2014, 2015), and Krechmer et al. (2016)). It should be noted the literature results were analyzed with fixed area to volume ratio of 3 m⁻¹ and fixed $C_{FEP\_surface}$ of 32.2 mg m⁻³ (Huang et al., 2018b).*" (Line 211)

"*Figure 6. Activity coefficient ($\gamma_{FEP\_surface}$) of organic compounds in FEP film. The sources of data include this work and the literature (compiled by Huang et al. (2018b), including Matsunaga and Ziemann (2010), Yeh and Ziemann (2014, 2015), and Krechmer et al. (2016)) (Matsunaga and Ziemann, 2010; Yeh and Ziemann, 2014, 2015; Krechmer et al., 2016).*" (Lines 411-412)

---

## Author Comment (AC3)

We appreciate the reviewer for providing us useful comments. In the following, original reviewer comments, author's responses, and corresponding updates on the main text are shown as purple, black, and *italic*. Line numbers in the responses correspond to those in the originally submitted version.

**Reviewer #3**

(3.1) Vapor-wall loss plays a pivotal role in smog chambers and should be considered when assessing atmospheric processes conducted in such environments. In their study, He et al. explored the impact of temperature on vapor-wall loss for n-alkanes within a smog chamber with Teflon walls. Their findings substantiated the hypothesis that vapor-wall loss becomes more significant at lower temperatures in a Teflon-walled chamber. This research has resulted in empirical equations that enhance the practicality of data analysis for chamber experiments. The experimental protocol is carefully designed, and the procedure for determining partition, desorption, and diffusion rate coefficients seems rational. I have just two primary concerns before the paper is accepted with some minor revisions.

We thank the reviewer for the positive comments. Responses to the individual inputs are given below.

(3.2) Typically, a nebulizer is employed to produce aerosols. While the 11-D Grimm OPC detected relatively low particle levels in the chamber, it is helpful to cross-verify these results using alternative particle sizing instruments, such as a scanning mobility particle sizer. This precaution is necessary because the OPC has a higher particle size detection limit (>200 nm), and following evaporation of hexane, nanoparticles might persist, potentially distorting the measurement of vapor concentration with TAG.

We thank the reviewer for the valuable input. Unfortunately, scanning mobility particle sizer was not employed in our experiment. Nevertheless, we carefully designed the injection concentration for the alkane to be lower than 20% of its saturation concentration under the corresponding experimental temperature. The first SV-TAG measurement was conducted at least 1 h after injection in each experiment, providing sufficient time for both the solvent and alkanes to evaporate. Even if some nanoparticles might persist, it is expected to only account for a small mass fraction of alkanes in the chamber air. We revised the text to point out this issue clearly:

"*The resulting initial concentrations ($C_0$) of individual n-alkanes in the chamber ranged from 4 to 50 $\mu g\ m^{-3}$ assuming no wall loss, which were lower than 20% of their saturation concentrations under the corresponding experimental temperature to avoid particle formation.*" (Lines 81-82)

(3.3) My second concern is about the vapor-wall surface interaction mechanism at lower temperatures, as also pointed out by Reviewer 2. At lower temperatures, vapor molecules exhibit a tendency to remain in the condensed phase. This phenomenon was observed by the measurements of vapor-wall interactions conducted in this study. There seems to be a discrepancy between Cw and gamma specifically at these lower temperatures. It would be advantageous to establish a self-consistent vapor-wall interaction mechanism, as this would prove beneficial to both readers and the Teflon-walled chamber user community.

We thank the reviewer for the valuable input. As previously mentioned in responses to comment (2.2) and (2.3), we provided an extra comparison with pre-cooled chamber results and demonstrated that our results were not path-dependent. Consistent slopes in Fig. 5 indicated C_FEP_Surface for alkanes would be constant under different temperatures. To clarify, we revised the main text and supporting information as below:

In main text:

"*For experiments below room temperature, the cooling system of the freezer was turned on one hour after the completion of the injection. The operation procedure was employed to avoid homogeneous nucleation and subsequent condensational growth of aerosol particles. The validity of employing this post-cooled operation procedure was demonstrated by comparison with a pre-cooled chamber result at 270 K (Figure S2).*" (Lines 83-85)

Added in supporting information:

[Figure]

Figure S2. Comparison of $C_{gas}/C_0$ between pre-cooled experiment at 270 K (hollow points) and post-cooled experiment at 270.2 K (solid points). The two-layer kinetic sorption model was employed to fit the pre-cooled chamber data (black solid lines). Results in post-cooled experiments (solid points) aligns well with the two-layer kinetic model fitting results of pre-cooled experiment (black lines). The consistency between pre- and post- cooled experiment demonstrated the validity of employing post-cooled operation procedure.

---

## Author Response (AR2)

We appreciate the reviewer for providing us useful comments. In the following, original reviewer comments, author's responses, and corresponding updates on the main text are shown as purple, black, and *italic*. Line numbers in the responses correspond to those in last submitted version.

(1.1) Overall the authors have responded to the referee comments very well. I have just one serious concern, which is that the discussion, data and analysis around uncertainties in using 3 hour data for Csurface/Cgas is confusing and lacking (detailed below). I recommend publication subject to this minor revision.

We thank the reviewer for the positive comment on our last reply. Response to the specific comment is given below.

(1.2) Specifically, Text S1 indicates that uncertainties are potentially major. This needs stating and discussing (including implications for results) explicitly in the results section, rather than referring readers to the supplementary section without further discussion. In addition, Text S1 makes very little sense to me: Figure S4b does not exist. I dont know what is overestimated by 7 and 55 %, and what the implications are for the main paper results. And the last two sentences of Text S1 makes no sense to me: what is the method that is doing the overestimating, what does employing 10 times mean, and what are the implications for main text results?

We thank the reviewer for this comment. The original content of Text S1 described the discrepancy between the $K_{eq}$ and our approximation of $K_{eq}$ as $1/[C_{gas}/C_0]_{\text{at 3 hours}} - 1$. We acknowledge the reviewer for pointing out that the figure number in the previous manuscript was inaccurate. The figure number was updated in the revised manuscript.

In the revised manuscript, we fully updated the corresponding descriptions, considering the reviewers' comment. Namely, we provided a detailed comparison of $K_{eq}$ and $1/[C_{gas}/C_0]_{\text{at 3 hours}} - 1$ for the room temperature experiments in Table S3. The employment of the room temperature data provides a quantitative comparison, as any temperature controlling processes were not needed. The experiments at room temperature were conducted for three times, allowing to estimate experimental uncertainties as standard deviations among the replicated runs. The result is summarized in Table S3 of the revised supplement file, demonstrating that the $K_{eq}$ and $1/[C_{gas}/C_0]_{\text{at 3 hours}} - 1$ agree well within experimental uncertainties. Overall, the approximation method, with an average overestimation by 22%, would not influence our main results. This result demonstrated the validity of employing $1/[C_{gas}/C_0]_{\text{at 3 hours}} - 1$. The corresponding descriptions were updated in the revised manuscript and supplement file, as detailed in the following.

In main text:

"" (Lines 177-178)

"*It is challenging to retrieve the value of $[C_{surface}/C_{gas}]_{eq}$ by fitting the data of the low-temperature experiments using the two-layer model, since the chamber was cooled after the injection of n-alkanes. Alternatively, the value of $[C_{surface}/C_{gas}]_{eq}$ was approximated using $1/[C_{gas}/C_0]_{\text{at 3 hours}} - 1$, assuming that diffusion of n-alkanes to the inner layer was still a minor loss process within 3 hours. Potential uncertainties associated with this approximation are summarized in Text S1. The uncertainties were estimated in two ways: (1) kinetic simulation based on fitting*

*parameters in Figure 3 (Figure S5) and (2) comparison of the retrieved values of $[C_{surface}/C_{gas}]_{eq}$ (i.e., $K_{eq}$) and*

$1/[C_{gas}/C_0]_{at\ 3\ hours} - 1$ *at room temperature (Table S3). The room-temperature experiments were conducted for three runs, allowing for the estimation of experimental uncertainties as standard deviation. Although the kinetic simulation implies overestimates of 7 - 55%, the measurement-based comparison demonstrates that* $1/$

$[C_{gas}/C_0]_{at\ 3\ hours} - 1$ *and* $[C_{surface}/C_{gas}]_{eq}$ *agreed within the experimental uncertainties, thereby supporting the*

*validity of the approximation.*" (Lines 199-201)

In supplement:

We revised the title of Figure S5 to clarify:

"*(b) Time series of the ratio of mass in surface and gas phase ($C_{surface}/C_{gas}$) and the ratio of mass not in and in the gas phase ($1/[C_{gas}/C_0] - 1$) for $C_{14}$ and $C_{19}$ n-alkanes. Solid and dashed lines represent $C_{14}$ and $C_{19}$ n-alkanes respectively. Red and blue lines represent the values of $C_{surface}/C_{gas}$ and $1/[C_{gas}/C_0] - 1$  respectively. The gray solid line indicates 3 hours as we choose the 3-hour measurements ($1/[C_{gas}/C_0]_{at\ 3\ hours} - 1$) to  approximate $K_{eq}$ in the main text.*"

Also, we added a comparison in Table S3:

"Table S3. Fitting and measurement results for room-temperature experiments.

| Compound | Experiment shown in Figure 3[a] | | | All three room-temperature experiments[b] | | |
|---|---|---|---|---|---|---|
| | $k_1$ (s$^{-1}$) | $k_{-1}$ (s$^{-1}$) | $k_2$ (s$^{-1}$)[c] | $K_{eq}$ | $1/[C_{gas}/C_0]_{at\ 3\ hours} - 1$ | relative difference |
| C$_{14}$ n-alkane | $2.76 \times 10^{-4}$ | $2.07 \times 10^{-4}$ | $9.39 \times 10^{-6}$ | $1.00 \pm 0.27$ | $1.14 \pm 0.23$ | $16\% \pm 8\%$ |
| C$_{15}$ n-alkane | $2.79 \times 10^{-4}$ | $2.07 \times 10^{-4}$ | $1.87 \times 10^{-5}$ | $1.08 \pm 0.19$ | $1.23 \pm 0.21$ | $13\% \pm 6\%$ |
| C$_{16}$ n-alkane | $3.41 \times 10^{-4}$ | $1.84 \times 10^{-4}$ | $2.67 \times 10^{-5}$ | $1.72 \pm 0.17$ | $1.98 \pm 0.14$ | $16\% \pm 11\%$ |
| C$_{17}$ n-alkane | $4.63 \times 10^{-4}$ | $1.82 \times 10^{-4}$ | $3.60 \times 10^{-5}$ | $2.73 \pm 0.50$ | $3.27 \pm 0.15$ | $23\% \pm 16\%$ |
| C$_{18}$ n-alkane | $6.35 \times 10^{-4}$ | $2.13 \times 10^{-4}$ | $4.35 \times 10^{-5}$ | $3.91 \pm 1.27$ | $4.85 \pm 0.57$ | $31\% \pm 23\%$ |
| C$_{19}$ n-alkane | $1.17 \times 10^{-3}$ | $2.25 \times 10^{-4}$ | $4.62 \times 10^{-5}$ | $8.69 \pm 4.37$ | $10.53 \pm 2.80$ | $35\% \pm 27\%$ |

[a] Optimized parameter sets of the two-layer model used in Figure 3. Best-fit parameters were obtained by the Newton method via Wolfram Mathematica 13.1.

[b] (Mean value) ± (standard deviation) are presented.

[c] $k_2$ obtained here was not used in Section 3.3 (characterization of diffusion in the Teflon wall). These fittings overestimated the first-order loss rate constant for low volatile species, C$_{16}$ – C$_{19}$ n-alkanes, as shown in Figure 3."

And we fully revised Text S1:

"*Text S1. Uncertainty in approximating $K_{eq}$ as $1/[C_{gas}/C_0]_{at\ 3\ hours} - 1$ *"

"*The uncertainties in approximating $K_{eq}$ as $1/[C_{gas}/C_0]_{at\ 3\ hours} - 1$ were estimated in two ways. First, the fitting parameters for room-temperature experiment in Figure 3 were used to simulate the kinetic process of wall loss (Figure S5a). Specifically, the values of $C_{surface}/C_{gas}$ and $1/[C_{gas}/C_0] - 1$ for $C_{14}$ and $C_{19}$ n-alkanes were retrieved, which were shown as red and blue lines in Figure S5b. These two n-alkanes were chosen, as they represent the highest and lowest volatile species in the room-temperature experiment. For both n-alkanes, $C_{surface}/C_{gas}$ stabilized by 3 hours, suggesting gas-surface partitioning reached equilibrium. In other words, $K_{eq}$ equals to*

$C_{surface}/C_{gas}$ at 3 hours. The discrepancy between the red lines and corresponding blue lines at 3 hours in Figure S5b was thus the bias caused by approximating $K_{eq}$ as $1/[C_{gas}/C_0]_{at\ 3\ hours} - 1$. For C$_{14}$ and C$_{19}$ n-alkanes, this approximation overestimated the values of $K_{eq}$ by 7% and 55%, respectively.

The uncertainties were also estimated by the room-temperature experimental results. Since no temperature controlling process was needed for room temperature experiments, the experimental data can be fitted using the two-layer model. Values of $k_1$ and $k_{-1}$, and thus $K_{eq}$ $(=k_1/k_{-1})$ can be obtained. Three sets of experiments were conducted at room temperature, which allows for estimating the experimental uncertainties as standard deviations among these replicated runs. As shown in Table S3, values of $K_{eq}$ and $1/[C_{gas}/C_0]_{at\ 3\ hours} - 1$ agree well within experimental uncertainties. On average, this approximation overestimates $K_{eq}$ by 22%, which would not influence our main results.

In summary, although the model simulation implies potentially large uncertainties, the comparison based on experimental data demonstrates the validity of the approximation. ~~Uncertainties in $C_{surface}/C_{gas}$ for C$_{14}$ and C$_{19}$ n-alkanes at room temperature by employment of the 3-hour measurement data are shown in Figure S4b. The discrepancy between the red lines and corresponding blue lines at 3 hours is the bias caused by the employment of 3-hour measurements. For C$_{14}$ and C$_{19}$ n-alkanes, this method overestimates 7% and 55 %. A more extreme case was used to simulate the low-temperature experiment by employing 10 times at k$_1$ of C$_{19}$ n-alkane and the maximum k$_2$ in all the experiments. In this case, this method overestimates $C_{surface}/C_{gas}$ within 62 % of the theoretical values.~~"